# The Impact of Grounding in Running Shoes on Indices of Performance in Elite Competitive Athletes

**DOI:** 10.3390/ijerph19031317

**Published:** 2022-01-25

**Authors:** Borja Muniz-Pardos, Irina Zelenkova, Alex Gonzalez-Aguero, Melanie Knopp, Toni Boitz, Martin Graham, Daniel Ruiz, Jose A. Casajus, Yannis P. Pitsiladis

**Affiliations:** 1Faculty of Health and Sports Science (FCSD), Department of Physiatry and Nursing, University of Zaragoza, 50009 Zaragoza, Spain; bmuniz@unizar.es (B.M.-P.); alexgonz@unizar.es (A.G.-A.); 2GENUD (Growth, Exercise, Nutrition and Development) Research Group, Department of Physiatry and Nursing, University of Zaragoza, 50009 Zaragoza, Spain; iz@i1.ru (I.Z.); joseant@unizar.es (J.A.C.); 3International Federation of Sports Medicine (FIMS), 1007 Lausanne, Switzerland; 4adidas Innovation, adidas AG, 91074 Herzogenaurach, Germany; Melanie.Knopp@adidas.com (M.K.); Toni.Boitz@adidas.com (T.B.); martin.william.graham@adidas.com (M.G.); daniel.ruiz@adidas.com (D.R.); 5Faculty of Medicine, Department of Physiatry and Nursing, University of Zaragoza, 50009 Zaragoza, Spain; 6School of Sport and Health Sciences, University of Brighton, Eastbourne BN20 7SN, UK; 7Centre for Exercise Sciences and Sports Medicine, FIMS Collaborating Centre of Sports Medicine, University of Rome “Foro Italico”, 00135 Rome, Italy; 8European Federation of Sports Medicine Associations (EFSMA), 1007 Lausanne, Switzerland

**Keywords:** earthing, environmental physiology, running performance, running economy, shoe technology, grounding

## Abstract

The introduction of carbon fiber plate shoes has triggered a plethora of world records in running, which has encouraged shoe industries to produce novel shoe designs to enhance running performance, including shoes containing conductor elements or “grounding shoes” (GS), which could potentially reduce the energy cost of running. The aim of this study was to examine the physiological and perceptual responses of athletes subjected to grounding shoes during running. Ten elite runners were recruited. Firstly, the athletes performed an incremental running test for VO_2_max and anaerobic threshold (AT) determination, and were familiarized with the two shoe conditions (traditional training shoe (TTS) and GS, the latter containing a conductor element under the insole). One week apart, athletes performed running economy tests (20 min run at 80% of the AT) on a 400 m dirt track, with shoe conditions randomized. VO_2_, heart rate, lactate, and perceived fatigue were registered throughout the experiment. No differences in any of the physiological or perceptual variables were identified between shoe conditions, with an equal running economy in both TTS and GS (51.1 ± 4.2 vs. 50.9 ± 5.1 mL kg^−1^ min^−1^, respectively). Our results suggest that a grounding stimulus does not improve the energy cost of running, or the physiological/perceptual responses of elite athletes.

## 1. Introduction

During the past five years, shoe designs have experienced a great technological revolution, which has been accompanied by a plethora of world records in all long-distance running events (i.e., from 5000 m to marathons, in both male and female athletes). Joyner et al., recently suggested that the factors potentially explaining the recent records in long-distance running are the physiological and training factors, in addition to shoe technology and drafting [1]. However, the abrupt drop in world records across all distances since 2017 suggests that shoe technology has a major contribution when compared to the other factors (i.e., training methods, the physiology of athletes, and drafting are factors that have not substantially changed in the last 5 years) [2].

The most popular shoe technology for road running includes a carbon fiber plate (CFP) within the sole, a light and highly reactive foam, and a stack up to 40 mm in thickness. This technology has been shown to reduce the energy cost of running during a fixed exercise intensity (traditionally between 14 and 18 km h^−1^) by approximately 4%, when compared to non-CFP shoes [3,4,5]. This improved running economy (RE) seems to be elicited by an increase in energy return caused by the action of passive elastic recoil, which in turn increases stride length and contact times, reduces step frequencies, and slightly increases the peak forces upon ground contact, when compared to non-CFP shoes [3,6,7].

The great popularity and effectiveness of CFP shoes has encouraged the shoe industry to explore new forms of shoe designs to optimize both health and performance during running. The implementation of “grounding” in humans purports to take advantage of the prolonged contact between an individual and the ground, and the potential transmission of energy between the two. Previous research states that the “direct contact of humans with the earth or using a metal conductor changes the electric potential on the surface of the body, as well as within the entire human organism” [8]. While the etiology of this potential effect is difficult to explain from a biophysiological perspective, previous findings have shown that the direct contact of an individual with the ground may reduce inflammatory processes, mood, pain, and stress at rest [9,10,11] and during exercise [8,9], with some studies suggesting that grounding technology may have a medical application. For example, previous research has suggested that the implementation of grounding is beneficial for mood, and may be especially beneficial in cases of depression, anxiety, stress, and trauma [11,12].

In relation to the existing research on grounding and exercise, an informative pilot study examined the effects of grounding on muscle physiology in response to exercise-induced muscle damage, and observed faster muscle recovery times under the grounding condition compared to the placebo [13]. The same group performed a more comprehensive follow-up study [14], observing that grounding significantly reduced creatine kinase (CK) levels 24 h post-exercise when compared to the placebo, suggesting that grounding may reduce acute muscular damage post-exercise. Following these early studies on grounding and muscle damage, a further study focused on the impact that this technology may have during aerobic exercise [8]. Sokal et al. claimed that the indirect contact of cyclists with the ground (through a metal conductor) while exercising elicited an increase in the electrical potential of the body when compared to those in the control group (not grounded). This study further reported that the observed increase in electrical potential with was accompanied by a greater decrease in blood urea concentrations during and after a 30 min cycling test at 50% of VO_2_max, indicating, according to the authors, a decreased physiological stress [8]. While these previous studies showed a benefit of grounding on the muscle recovery and physiological stress of healthy subjects in response to different modes of exercise (i.e., resistance training and cycling), the impact of this technology while running is unknown.

Given the imminent introduction of grounding technology in running shoes, and the absence of rigorous scientific evidence of its effects, adding conductor elements within the shoe and employing a well-controlled experimental design, would allow for the assessment of any putative effects of this technology (i.e., grounding technology in running shoes) during running. This is especially important given the recent controversy that novel shoe technologies are negatively impacting the integrity and fairness within sport [2,15]. A recent critical review [2] highlighted how novel shoe designs are revolutionizing the world of sport, as numerous National, European, World, and Olympic records have been broken over an extraordinarily short time period (i.e., since the introduction of CFP shoes). In addition to this controversy, there is a lack of well-controlled and rigorous studies in the field that focus on the impact of shoe designs on running performance [2], which makes the true performance benefits of certain shoe technologies difficult to determine.

Considering the reduced physiological stress and muscle damage witnessed in subjects while performing other physical activities (i.e., strength exercises and cycling), it is important to examine the impact of grounding on the physiological and perceptual responses to running, especially considering the interest of shoe companies in incorporating grounding technology into running shoes, and the potential fairness/integrity issues that may result if a performance benefit is demonstrated. Therefore, the main aim of the present study was to compare the RE and physiological stress of well-trained runners while running in either grounding shoes (GS) or traditional training shoes (TTS).

## 2. Materials and Methods

### 2.1. Participants

Ten highly-trained male runners (age = 27 ± 7 years; weight = 64.6 ± 6 kg; height = 176.3 ± 5.4 cm) were recruited for the present study. Upon recruitment, all subjects received and signed an informed consent form in order to participate in the study. Subjects were required to meet the following inclusion criteria: (1) to train a minimum of 50 km week^−1^, (2) to have a personal best under 35:00 min:s in 10 km or 17:30 min:s in 5 km, (3) to be healthy and without any musculoskeletal injury.

### 2.2. Procedures

The present study design required runners to visit either the laboratory or the track on two occasions, both separated by a period of 7 days to avoid any residual fatigue. Visit 1 included a VO_2_max test, ventilatory threshold determination, and shoe familiarization in the laboratory; Visit 2 included 20 min RE tests at 80% of the anaerobic threshold, on a 400 m dirt track, with the order of the two shoe conditions randomized (Figure 1). A dirt track was selected over a traditional synthetic PU rubber track to avoid any material interference between the ground and the athlete. The present study was approved by the Ethics Committee of Aragon (CEICA, num. 17/2021).

### 2.3. Shoe Conditions

Two shoe conditions were tested: the traditional training shoe (TTS) and the grounding shoe (GS), with these being visually identical as shown in Figure 1. Shoes with grounding potential contained a conductor element around the insole, and aimed to diminish the physiological stress experienced by the athlete during running as they run in closer contact with the ground. The insulation and thermal permeability of the shoes were considered similar, given that the same material was used for both experimental and non-experimental shoes, with the exception of the conductor element. Both uppers consisted of the same knitted textile, produced and supplied at the same time for both types of shoe (Figure 1). The GS upper included a textile webbing containing yarn that encouraged electrical charge to flow through the material. The material was stitched into the collar area, and ran through the midsole to connect with the rubber on the outsole that contacts the ground. The TTS outsole included conventional rubber, while the GS outsole included rubber that encouraged the flow of electrical charge. The manufacturers labelled the shoes with a number in red or blue according to the two shoe conditions, and this setting was used by the research team to keep the study design double-blinded (See Figure 1). Additionally, as each athlete may have become subjectively biased during the familiarization trial, all blue/red labels were obscured with tape in Visit 2. All athletes had their own pair of shoes for each shoe condition.

### 2.4. Visit 1. Maximal Oxygen Uptake and Ventilatory Threshold Determination

On the first day, athletes were subjected to a skin temperature test and a SARS-CoV-2 antigen test, in order to participate in this study. Upon testing negative, informed consent was signed by all participants, and medical history and pre-participation screening was also completed. The laboratory assessments performed during the first day included:

Anthropometric and body composition assessments. The parameters measured were as follows: weight, height, height from sitting position, foot length, calf circumference and fold, and thigh circumference and fold. Percent body fat, muscle mass, and bone mass were assessed with a DXA scan (Hologic Corp., Bedford, MA, USA). Body fat, body water, and muscle mass were also assessed via bioimpedance (TANITA BC 780-S MA, Tanita Corp., Tokyo, Japan).

Maximal aerobic capacity test. All subjects were previously familiarized with VO_2_max testing. Prior to the VO_2_max test, subjects laid down for 5 min, and resting electrocardiograms and blood pressure tests were performed and assessed by experienced medical doctors to ensure athletes did not have any cardiological issues. Participants breathed through a low dead space mask, with air sampled at 60 mL min^−1^. Before each test, two-point calibrations of the gas sensors were completed, using a known gas mixture (16% O_2_ and 5% CO_2_) and ambient air. Ventilatory volume was calibrated using a 3 L (±0.4%) syringe. Firstly, subjects performed a self-paced warm-up, and prior to the commencement of the test, subjects were instrumented with a portable metabolic analyzer (Cosmed K5, Cosmed Srl, Rome, Italy) and a heart rate device (Polar H10, Polar Electro, Kempele, Finland). A short-ramp incremental protocol was used (i.e., 13–16 min) as this has been shown to be the most appropriate assessment for identifying individual physiological events in well-trained runners [16,17,18]. The protocol consisted of a 3 min run at 10 km h^−1^ and a 1% gradient on a treadmill (h/p/cosmos, Nussdorf—Traunstein, Germany), followed by increases of 1 km h^−1^ min^−1^ until volitional exhaustion. Heart rate was monitored throughout the test, and overall perception of effort (RPE) and specific RPE for the legs were registered immediately after the test. This test enabled the determination of VO_2_max (defined as the highest 30 s mean values obtained during the test) and individual anaerobic threshold (IAT), determined through visual assessment conducted by two experienced exercise physiologists. Each individual speed for subsequent shoe trials were determined at the 80% of the IAT velocity. This VO_2_max test involved the subjects’ preferred shoe, and served to objectively quantify individual running speed for subsequent RE trials (avoiding the impact of the slow component of oxygen uptake given the repeated square-wave design of the RE tests on the second visit). Visit 1 also involved the familiarization of the different running shoes during a light, 5 min run with each pair of shoes, in preparation for Visit 2.

### 2.5. Visit 2. Running Economy Tests

During the second visit, indices of performance, with particular focus on RE, were assessed for each shoe condition, determined on a 400 m dirt track. Air temperature and humidity were recorded at the beginning and end of the experimental sessions using a portable meteorological station, and all trials were performed either in the early morning or late evening to avoid extreme environmental conditions. Participants breathed through a low dead space mask, with air sampled at 60 mL min^−1^. Before each subject’s first trial, the portable metabolic analyzer was calibrated following the calibration procedures aforementioned. The shoe conditions were randomly assigned, and both runners and assessors were blinded to the shoe condition. Brand new socks were used for each RE trial to avoid excessive humidity within the shoe, as this could impact grounding effect. Body mass was measured before and after each test. Each runner warmed up for 15 min with their preferred training shoes prior to being equipped with the portable metabolic analyzer. Pre-trial blood lactate was measured from a single drop of whole blood from the fingertip using a lactate meter (Lactate Pro 2, Arkray Europe, B.V., Amstelveen, the Netherlands), and pre-trial heart rate and RPE were also collected. Athletes performed two 20 min exercise bouts at 80% of their IAT velocity for each shoe condition, with a 20 min rest in between (Figure 2). The duration of this RE protocol was longer than traditional RE tests (4–6 min) used in previous studies examining shoe designs [3,4,5]. The reason for this was to allow for a longer contact time between the athlete and the earth, which is crucial for obtaining a dose–response relationship. Lactate, whole-body RPE, and legs-only RPE (1–10 scale) were recorded at min 1, 3, and 15 of recovery following both trials, and heart rate and ventilatory parameters were monitored throughout the test. A researcher (and experienced cyclist) paced all runners at their individual speed using a bicycle. The RE elicited by each shoe condition was determined as the mean VO_2_ between min 10 to min 15, as steady state was ensured during this period. To reduce the noise in the ventilatory measurements, a 7-breath averaging method was performed.

### 2.6. Statistical Analysis

Means and standard deviations (mean ± SD) were calculated for all variables. An a priori sample size calculation (G*Power software, version 3.1.9.3, Heinrich-Heine-Universität Düsseldorf, Düsseldorf, Germany) was performed using the running economy data reported in a previous study testing different shoe designs in well-trained athletes (Barnes et al., 2018). The VO_2_ data for both the control and grounded shoe (53.61 ± 2.20 vs. 51.26 ± 2.23 mL kg^−1^ min^−1^, respectively) were used to generate a correlation coefficient of 0.45 and a Cohen’s *d* of 1.01. A two-tailed *t*-test revealed that a total sample size of 10 subjects was required to obtain statistical power of 0.80 and an alpha of 0.05. A Shapiro–Wilk test revealed normal data distributions across all studied variables. Student’s *t*-tests for paired samples were applied between TTS and GS shoe conditions in order to examine the differences between metabolic and RE data (HR, VO_2_, RER). Significant values were set at *p* ≤ 0.05 and effect sizes (Cohen’s *d*) were also calculated. The Statistical Package for the Social Sciences (SPSS) version 23.0 (SPSS Inc., Chicago, IL, USA) was used to perform the statistical analyses.

## 3. Results

A final sample of 10 athletes completed the present study, with no drop-outs. These athletes were national to international level runners/triathletes, with two of them having participated in major sporting events (Olympic Games and World Championships). Table 1 presents the mean and individual descriptive characteristics of the sample, showing a fairly homogeneous fitness level across all runners (i.e., mean VO_2_max of 78.4 ± 3.8 mL kg^−1^ min^−1^).

A Student’s t-test for paired samples revealed no significant difference in RE values between TTS and GS conditions (51.1 ± 4.2 vs. 50.9 ± 5.1 mL kg^−1^ min^−1^, respectively, *p* = 0.779, Cohen’s *d* = 0.092). Figure 3 shows both mean and individual values for VO_2_. Additionally, blood lactate was not different between shoe conditions at min 1 (*p* = 0.793), min 3 (*p* = 0.250), and min 15 (*p* = 0.641) post-exercise (Figure 4). Both whole-body and legs-only RPE values were also not significantly different between TTS and GS at min 1 (*p* = 1.0 and *p* = 0.273, respectively), min 3 (*p* = 0.443 and *p* = 0.591, respectively), and min 15 (*p* = 0.168 and *p* = 0.591, respectively) post-exercise (Figure 4). Finally, HR values were not significantly different between TTS and GS during exercise (150.1 ± 15 vs. 151.0 ± 16 bpm, respectively, *p* = 0.461, Cohen’s *d* = 0.244; Figure 4).

## 4. Discussion

The main findings of the present study show that grounding technology applied to shoe designs does not provide a physiological/perceptual response over traditional training shoes in well-trained athletes. The RE, blood lactate, heart rate, and perceptual response of these athletes, exercising at 80% of their IAT during 20 min on a 400 m dirt track, were not different between shoes conditions.

Despite previous promising findings suggesting that grounding technology has positive effects on the physiological responses (i.e., reduced acute inflammatory processes) of humans at rest [7,8], very limited research has focused on the implementation of grounding during exercise, with only two studies focusing on the effectiveness of grounding in reducing muscular damage after exercise-induced DOMS. This is the first study to examine the impact of grounding in shoes during running, which makes the comparison with previous studies challenging due to the unique nature of running for the implementation of this technology (i.e., intermittent contact time with the ground). Our findings, however, differ from those of Sokal et al. [8], who claimed that all recreational cyclists within their study experienced physiological attenuation at rest, during a 30 min exercise at 50% of their VO_2_max, and during recovery, indicated by decreases in blood urea; however, these authors failed to include any individual data. It is also worth noting that these biochemical parameters were not measured immediately prior to grounding/placebo conditions, and therefore group-by-time interactions could not be determined, which limits the interpretation of these results. Additionally, one would expect both blood urea and creatinine concentrations to remain unchanged following the exercise protocol used by these authors (a single bout of light exercise for 30 min). Blood urea and creatinine levels have been shown to increase after prolonged, strenuous exercise as a result of increased protein catabolism and/or impaired renal function [19], which is unlikely to have occurred during the exercise protocol proposed by Sokal et al. The difference between the groups observed by Sokal et al., interpreted in the context of our present findings, are more likely due to day-to-day inter-individual variability in blood urea, or some potential methodological issues during data collection, rather than due to physiological stress attenuation during exercise. In a subsequent study, Sokal et al. presented additional data from the same aforementioned experiment [20], focusing on the effects of grounding on VO_2_ uptake, blood glucose, lactate, and bilirubin concentrations. The 42 subjects included in this study were divided into two subgroups (*n* = 21) according to their VO_2_max, therefore, both groups had a comparable cardiorespiratory fitness (Group A = 50.8 vs. Group B = 50.7 mL kg^−1^ min^−1^). The study design followed a double-blind, crossover protocol between Groups A and B. During the first testing day, Group A was under the placebo condition and Group B was under the grounding stimulus, with these conditions interchanged during the second day of testing. These authors reported a significantly reduced VO_2_ uptake (numeric data not shown by the authors) at the end of the exercise with the grounding stimulus only in Group B, when compared to the placebo. The study design employed by Sokal et al. [8,20] has limited reliability, given that their experimental tests were performed on different days, which may have biased the results. Day-to-day variability and the lack of a familiarization trial may have potentiated the learning effects only for Group B (i.e., the group with the grounding stimulus during the second day). These results should, therefore, be interpreted with caution.

To our knowledge, the two aforementioned studies are the only two experiments focusing on the effects of grounding on the biophysiological responses of humans during submaximal exercise. However, the important methodological issues described above, and the use of cycling being the only mode of exercise, limits the interpretation of the current literature and its comparison with the present study. In our experiment, we used a double-blind, randomized, crossover design, with tests for all experimental conditions performed on the same day. We are aware that the conductor element within the shoe was not in permanent contact with the ground (i.e., intermittent contact time during running), and we did not measure muscle activity, nor foot/stride mechanics, during running, which may have provided more information and potentially revealed an effect. However, to ensure a sufficient contact time, we designed a longer than usual RE protocol (i.e., 20 min bouts; Figure 2), so that we could identify a potential dose–response relationship over time. Despite these rigorous experimental procedures, our results show that grounding technology did not have any impact on the measured responses during running when compared to traditional training shoes. Previous research showed a decrease in muscle damage in response to high-intensity strength exercises in subjects under grounding conditions [13,14] when compared to a placebo. These findings would suggest that grounding technology may have a role to play as a muscle recovery method, which in turn could translate into a benefit for runners when performing higher intensity exercise (i.e., above the anaerobic threshold) in which muscle fatigue and acidosis occur to a greater extent. Nonetheless, future research using larger sample sizes and examining foot mechanics (especially contact times) would be required to confirm our findings. Other shoe designs currently available on the market that include a CFP and a high midsole stack height made of compliant, resilient, and lightweight foam, seem the most effective shoe modality to date. This technology has shown to improve RE by increasing the midsole longitudinal bending stiffness, favoring a decrease in the range of motion of the metatarsophalangeal joint [3,21,22].

## 5. Conclusions

In conclusion, our results suggest that grounding in shoe designs is not an effective alternative for well-trained athletes to improve their running efficiencies, and/or their physiological/perceptual responses during submaximal exercise. However, there are intrinsic limitations that should be considered. Potential grounding effects could have been missed during our study as running does not allow constant contact between the athlete and the ground, which could have potentially biased the results. In relation to this, lower caliber athletes may have benefited from this technology given their ground contact times are greater than faster, elite athletes; an issue that could not be addressed in the current study. Future research may therefore consider additional sports in which athletes remain in constant contact with the ground (e.g., race-walking, cross-country skiing, powerlifting). Despite these limitations, our study followed a high-quality methodological protocol (double-blind, randomized, crossover design) using a homogeneous sample of highly trained athletes (as represented in Table 1), which suggests that our conclusions are reliable for this specific population.

## Figures and Tables

**Figure 1 ijerph-19-01317-f001:**
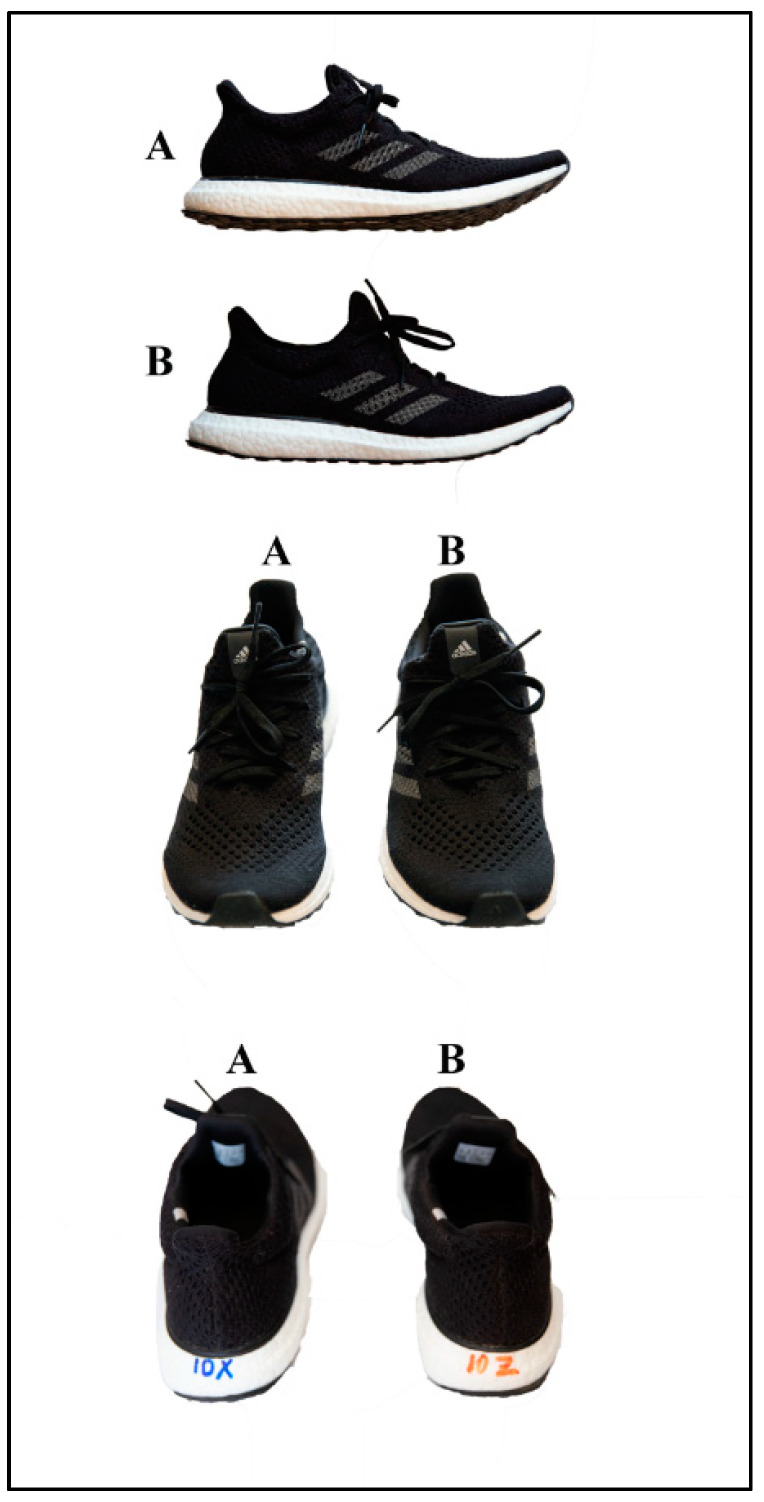
Image of the right grounding shoe (**A**) and traditional training shoe (**B**) for one of the elite athletes.

**Figure 2 ijerph-19-01317-f002:**
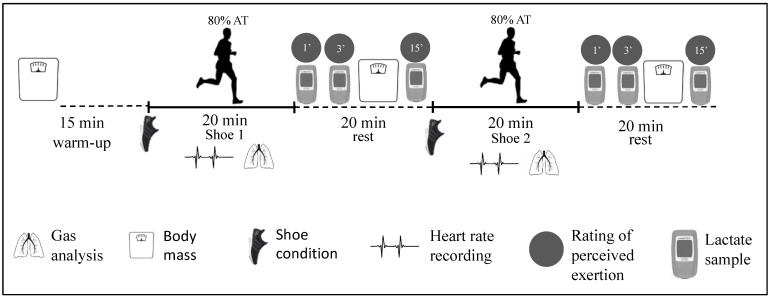
Protocol for the running economy trials at 80% of the anaerobic threshold (AT).

**Figure 3 ijerph-19-01317-f003:**
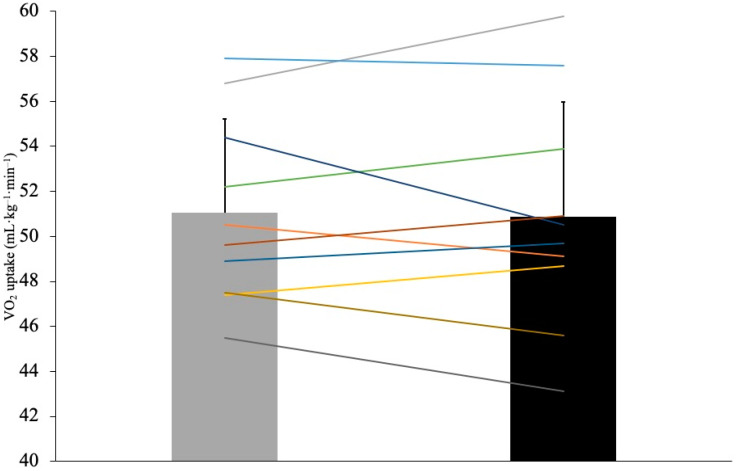
Mean and individual running economy values (mL kg^−1^ min^−1^) of the 10 athletes running in traditional training shoes (grey column) and in grounding shoes (black column).

**Figure 4 ijerph-19-01317-f004:**
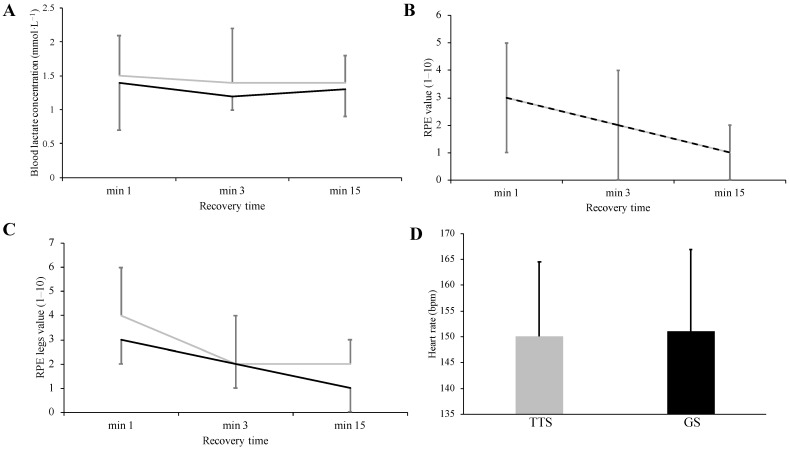
Blood lactate (**A**), whole-body rate of perceived exertion (RPE; **B**), and legs-only RPE (**C**) during the recovery period after running in the traditional training shoe (TTS, gray solid line) or grounding shoe (GS, black solid line). Heart rate during the running economy trial in both TTS and GS trials (**D**). Dashed lines represent overlapping mean values between shoes.

**Table 1 ijerph-19-01317-t001:** Descriptive characteristics of the participants.

ID	Age (years)	Weight (kg)	Height (cm)	BMI (kg m^−2^)	Bioimpedance (Fat %)	VO_2_max (mL kg^−1^ min^−1^)
Athlete 1	31.0	78.5	180.3	24.1	12.7	76.0
Athlete 2	25.7	65.7	177.8	20.8	5.5	82.3
Athlete 3	35.0	64	174.3	21.1	10.4	80.3
Athlete 4	20.8	68.9	186.3	19.9	11.8	83.6
Athlete 5	31.1	57.0	171.0	19.5	3.0	78.0
Athlete 6	26.2	59.3	170.2	20.5	11.2	77.8
Athlete 7	38.2	66.0	176.5	21.2	3.8	78.5
Athlete 8	25.0	72.5	177.7	23.0	7.0	77.3
Athlete 9	20.6	64.9	171.2	22.1	8.9	80.5
Athlete 10	18.1	64.0	183.0	19.1	8.5	69.9
Mean ± SD	27.2 ± 6.6	66.1 ± 6.2	176.8 ± 5.4	21.1 ± 1.6	8.3 ± 3.4	78.4 ± 3.8

## Data Availability

The datasets used and analyzed within the present manuscript will be available from the corresponding author/first author upon request.

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
