# Peer review of "The Impact of Grounding in Running Shoes on Indices of Performance in Elite Competitive Athletes"

_ijerph, 2022, doi:10.3390/ijerph19031317_

Round 1

Reviewer 1 Report

Review Summary Report IJERPH

Article title:

The impact of grounding technology in running shoes on indices of performance in highly-trained individuals

I would rethink the name of the article. It might be of interest to qualify the study as a pilot study.

Abstract

Is presented properly

Introduction

Line 47, “reduces peak forces upon ground contact”, there are some studies that conclude the opposite, please check the first paper of Hoogkamer et al, where the vertical peak is tackled.

Line 51. Some papers conclude that the grounding running technique (high duty factor) has several advantages for recreational runners. In this line, the grounding technique and technology is mixed. I suggest the authors to present both concepts separately.

Considering the novelty of “grounding technology shoes” I would suggest the authors to explain the technology and evidence behind it. Please bear in mind that, since it is the first study in running you must present good argumentation in support of the use of this approach in running footwear. This is paramount to consider your paper suitable for publication in this journal. The information included in the manuscript is not enough so far. The state of the art of earthing states that this method has positive effects related to pain reduction, stress, mood and impact in chronic health disorders (Menigoz et al 2019), it would be interesting to know the effects on some variables related to physical performance.

Since there are no studies focused in running I suggest to justify very well the hypothesis of this study and why it is needed.

Methods

The methods are properly presented.

Considering the sample size, I would present the statistical power measured post study. Moreover, please explain why you have chosen N=10 as the experimental sample for your study.

Considering the importance of the ground contact time for this technology and study, I miss in the study some important variables such as the duty factor and cadence. I suggest the authors to enrich the study with this analysis.

Results

Results are correctly presented. However, I missed relevant variables.

Discussion

Since there is no other study in running the discussion was done using mainly three references, one of them not related to sports, the other two focused on cycling.

Question: Did you find any study related to human gait?

The discussion discusses several interesting variables that have not been analysed in this study, making it difficult to assess the contribution of this study to scientific knowledge.

The sample size of this study is one important limitation, it should be tackled in the discussion.

The number of references is very low considering the quality of this journal (Q1), I suggest adding more relevant references specially in the introduction.

Conclusion

No comment

Reviewer 2 Report

Manuscript ID ijerph-1501087

Title The impact of grounding technology in running shoes on indices of performance in highly-trained individuals

Article entitled ’ The impact of grounding technology in running shoes on indices of performance in highly-trained individuals’ is written structurally and legibly correctly. The subject of running shoes is interesting, especially taking into account the authors' approach and the study of two types of shoes.

The literary introduction gives the reader a good overview of the topic. The authors could, however, be tempted to provide more data, as they only use 7 literature items. The research aim is clearly and clearly defined.

The authors deviated from the subject of methodology very well. The research group was well characterized and selected. Preparation for the study and the research protocol were described very well. However, the basic parameters of the footwear, such as insulation or thermal permeability, are missing. Maybe it is worth adding them and looking at the results also from this perspective?

Results and discuusion are correct. Only authors should change figure 2 – it should be one chart, not two separate.

The conclusions were drawn legally taking into account the obtained results. The authors are aware of the limitations of the study, but the results are interesting, allowing the material to be published.

Reviewer 3 Report

Thank you for the opportunity to review this manuscript. 

The topic of the paper is interesting and fits the scope of the journal. The text is relatively well written and composed.

Please find below minor comments for improving the article.

Line 66. In the aim of this study, I would like to add about the comparison between the GS and TTS on RE.

Line 107. Please refer the reference that used for maximal aerobic capacity test.

Line 130. Please refer the reference that used for RE test.

Line 86. Please refer details about the characteristics of traditional shoes. For example, is the shoes that athletes use in daily training?

Round 2

Reviewer 1 Report

Comment 1.

After considering your reply, doing some calculation in Gpower,and reading the Barnes study (N=24) I strongly recommend to add in your title: “pilot study”.

Please link this answer with the answer of comment 8.

Comment 8.

I perform the Gpower a priori sample size calculation and I didn`t achieve your numbers. Further more, a Cohend`s D of 1.01 is a very high value, I would use a lower one. I found a sample size of 10 with a Cohend’s D of 0.72, Alpha=0.05, beta=0.80.

On the other hand, the Barnes study included 24 runners, I did not find in their paper any reference to the a priori sample size.

I agree with you, I would not include the post power test calculation, nevertheless I would name your study as “pilot study”.

Considering my calculations in Gpower a sample size of 19-20 would meet the requirements of this study.

Comment 9

Considering that the grounding technology is directly related to the contact with the ground, at least the contact time and the relation between contact and flight times are very important in this study. Please bear this in mind for future studies.

Comment 13

Having in mind my previous comments and the authors replies, I believe the sample size is a limitation of your study.

I hope you concord with me, please address it in the discussion.

Additional comment

Considering the novelty of the technology, please add a figure/picture of the shoe, that shows graphically this technology. It would add more value to your paper.
